# Efficient Training on Very Large Corpora via Gramian Estimation

**Walid Krichene, Nicolas Mayoraz, Steffen Rendle, Li Zhang, Xinyang Yi, Lichan Hong, Ed H. Chi & John Anderson**
Google
{walidk,nmayoraz,srendle,liqzhang,xinyang,lichan,edchi,janders}@google.com

## Abstract

We study the problem of learning similarity functions over very large corpora using neural network embedding models. These models are typically trained using SGD with random sampling of unobserved pairs, with a sample size that grows quadratically with the corpus size, making it expensive to scale. We propose new efficient methods to train such models without sampling unobserved pairs. Inspired by matrix factorization, our approach relies on adding a global quadratic penalty and expressing this term as the inner-product of two Gramians. We show that the gradient of this term can be efficiently computed by maintaining estimates of the Gramians, and develop variance reduction schemes to improve the quality of the estimates. We conduct large-scale experiments that show a significant improvement in training time and generalization performance compared to sampling methods.

## 1 Introduction

We consider the problem of learning a similarity function $h : \mathcal{X} \times \mathcal{Y} \to \mathbb{R}$, that maps each pair of items, represented by their feature vectors $(x, y) \in \mathcal{X} \times \mathcal{Y}$, to a real number $h(x, y)$, representing their similarity. We will refer to $x$ and $y$ as the left and right feature vectors, respectively. Many problems can be cast in this form: In natural language processing, $x$ represents a context (e.g. a bag of words), $y$ represents a candidate word, and the target similarity measures the likelihood to observe $y$ in context $x$ (Mikolov et al., 2013; Pennington et al., 2014; Levy & Goldberg, 2014). In recommender systems, $x$ represents a user query, $y$ represents a candidate item, and the target similarity is a measure of relevance of item $y$ to query $x$, e.g. a movie rating (Agarwal & Chen, 2009), or the likelihood to watch a given movie (Hu et al., 2008; Rendle, 2010). Other applications include image similarity, where $x$ and $y$ are pixel-representations of images (Bromley et al., 1993; Chechik et al., 2010; Schroff et al., 2015), and network embedding models (Grover & Leskovec, 2016; Qiu et al., 2018), where $x$ and $y$ are nodes in a graph and the similarity is whether an edge connects them.

A popular approach to learning similarity functions is to train an embedding representation of each item, such that items with high similarity are mapped to vectors that are close in the embedding space. A common property of such problems is that only a small subset of all possible pairs $\mathcal{X} \times \mathcal{Y}$ is present in the training set, and those examples typically have high similarity. Training exclusively on observed examples has been demonstrated to yield poor generalization performance. Intuitively, when trained only on observed pairs, the model places the embedding of a given item close to similar items, but does not learn to place it far from dissimilar ones (Shazeer et al., 2016; Xin et al., 2017). Taking into account unobserved pairs is known to improve the embedding quality in many applications, including recommendation (Hu et al., 2008; Yu et al., 2017) and word analogy tasks (Shazeer et al., 2016). This is often achieved by adding a low-similarity prior on *all pairs*, which acts as a repulsive force between all embeddings. But because it involves a number of terms quadratic in the corpus size, this term is computationally intractable (except in the linear case), and it is typically optimized using sampling: for each observed pair in the training set, a set of random unobserved pairs is sampled and used to compute an estimate of the repulsive term. But as the corpus size increases, the quality of the estimates deteriorates unless the sample size is increased, which limits scalability.

In this paper, we address this issue by developing new methods to efficiently estimate the repulsive term, without sampling unobserved pairs. Our approach is inspired by matrix factorization models,

which correspond to the special case of linear embedding functions. They are typically trained using alternating least squares (Hu et al., 2008), or coordinate descent methods (Bayer et al., 2017), which circumvent the computational burden of the repulsive term by writing it as a matrix-inner-product of two Gramians, and computing the left Gramian before optimizing over the right embeddings, and vice-versa. Unfortunately, in non-linear embedding models, each update of the model parameters induces a simulateneous change in all embeddings, making it impractical to recompute the Gramians at each iteration. As a result, the Gramian formulation has been largely ignored in the non-linear setting, where models are instead trained using stochastic gradient methods with sampling of unobserved pairs, see Chen et al. (2016). Vincent et al. (2015) were, to our knowledge, the first to attempt leveraging the Gramian formulation in the non-linear case. They consider a model where *only one of the embedding functions is non-linear*, and show that the gradient can be computed efficiently in that case. Their result is remarkable in that it allows exact gradient computation, but this unfortunately does not generalize to the case where both embedding functions are non-linear.

**Contributions**   We propose new methods that leverage the Gramian formulation in the non-linear case, and that, unlike previous approaches, are efficient even when both left and right embeddings are non-linear. Our methods operate by maintaining stochastic estimates of the Gram matrices, and using different variance reduction schemes to improve the quality of the estimates. We perform several experiments that show these methods scale far better than traditional sampling approaches on very large corpora.

We start by reviewing preliminaries in Section 2, then derive the Gramian-based methods and analyze them in Section 3. We conduct large-scale experiments on the Wikipedia dataset in Section 4, and provide additional experiments in the appendix. All the proofs are deferred to Appendix A.

## 2   PRELIMINARIES

### 2.1   NOTATION AND PROBLEM FORMULATION

We consider models that consist of two embedding functions $u : \mathbb{R}^d \times \mathcal{X} \to \mathbb{R}^k$ and $v : \mathbb{R}^d \times \mathcal{Y} \to \mathbb{R}^k$, which map a parameter vector[1] $\theta \in \mathbb{R}^d$ and feature vectors $x, y$ to embeddings $u(\theta, x), v(\theta, y) \in \mathbb{R}^k$. The output of the model is the dot product[2] of the embeddings $h_\theta(x, y) = \langle u(\theta, x), v(\theta, y) \rangle$, where $\langle \cdot, \cdot \rangle$ denotes the usual inner-product on $\mathbb{R}^k$. Low-rank matrix factorization is a special case, in which the left and right embedding functions are linear in $x$ and $y$. Figure 1 illustrates a non-linear model, in which each embedding function is given by a feed-forward neural network.[3]

We denote the training set by $T = \{(x_i, y_i, s_i) \in \mathcal{X} \times \mathcal{Y} \times \mathbb{R}\}_{i \in \{1, \ldots, n\}}$, where $x_i, y_i$ are the feature vectors and $s_i$ is the target similarity for example $i$. To make notation more compact, we will use $u_i(\theta), v_i(\theta)$ as a shorthand for $u(\theta, x_i), v(\theta, y_i)$, respectively. As discussed in the introduction, we also assume that we are given a low-similarity prior $p_{ij} \in \mathbb{R}$ for all pairs $(i, j) \in \{1, \ldots, n\}^2$. Given a differentiable scalar loss function $\ell : \mathbb{R} \times \mathbb{R} \to \mathbb{R}$, the objective function is given by

$$\min_{\theta \in \mathbb{R}^d} \frac{1}{n} \sum_{i=1}^n \ell\left(\langle u_i(\theta), v_i(\theta) \rangle, s_i\right) + \frac{\lambda}{n^2} \sum_{i=1}^n \sum_{j=1}^n (\langle u_i(\theta), v_j(\theta) \rangle - p_{ij})^2, \tag{1}$$

where the first term measures the loss on observed data, the second term penalizes deviations from the prior, and $\lambda$ is a positive hyper-parameter that trades-off the two terms. To simplify the discussion, we will assume a uniform zero prior $p_{ij}$ as in (Hu et al., 2008), the general case is treated in Appendix B. To optimize this objective, existing methods rely on sampling to approximate the second term, and are usually referred to as negative sampling or candidate sampling, see Chen et al. (2016); Yu et al. (2017) for a survey. Due to the double sum in (1), the quality of the sampling estimates degrades as the corpus size increases, which can significantly increase training times. This can be alleviated by increasing the sample size, but does not scale to very large corpora.

---

[1] In many applications, it is desirable for the two embedding functions $u, v$ to share certain parameters, e.g. embeddings of categorical features common to left and right items; hence, we use the same $\theta$ for both.

[2] This also includes cosine similarity models when the embedding functions $u, v$ are normalized.

[3] We focus on models with this dot-product structure since they allow efficient retrieval: given $x \in \mathcal{X}$, finding items $y \in \mathcal{Y}$ with high similarity to $x$ reduces to a maximum inner-product search problem (MIPS), which can be approximated efficiently (Shrivastava & Li, 2014; Neyshabur & Srebro, 2015).

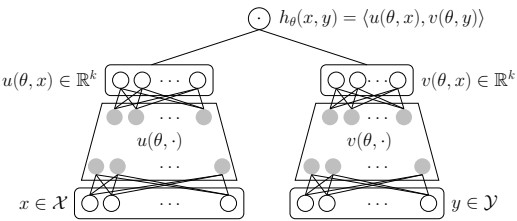

Figure 1: A dot-product embedding model for a similarity function on $\mathcal{X} \times \mathcal{Y}$.

## 2.2 GRAMIAN FORMULATION

A different approach to solving (1), widely popular in matrix factorization, is to rewrite the double sum as the inner product of two Gram matrices. Let us denote by $U_\theta \in \mathbb{R}^{n \times k}$ the matrix of all left embeddings such that $u_i(\theta)$ is the $i$-th row of $U_\theta$, and similarly for $V_\theta \in \mathbb{R}^{n \times k}$. Then denoting the matrix inner-product by $\langle A, B \rangle = \sum_{i,j} A_{ij} B_{ij}$, we can rewrite the double sum in (1) as:

$$g(\theta) \coloneqq \frac{1}{n^2} \sum_{i=1}^{n} \sum_{j=1}^{n} (U_\theta V_\theta^\top)_{ij}^2 = \frac{1}{n^2} \left\langle U_\theta V_\theta^\top, U_\theta V_\theta^\top \right\rangle. \tag{2}$$

Now, using the adjoint property of the inner product, we have $\left\langle U_\theta V_\theta^\top, U_\theta V_\theta^\top \right\rangle = \left\langle U_\theta^\top U_\theta, V_\theta^\top V_\theta \right\rangle$, and if we denote by $u \otimes u$ the outer product of a vector $u$ by itself, and define the Gram matrices

$$G_u(\theta) \coloneqq \frac{1}{n} U_\theta^\top U_\theta = \frac{1}{n} \sum_{i=1}^{n} u_i(\theta) \otimes u_i(\theta), \quad G_v(\theta) \coloneqq \frac{1}{n} V_\theta^\top V_\theta = \frac{1}{n} \sum_{i=1}^{n} v_i(\theta) \otimes v_i(\theta), \tag{3}$$

the penalty term becomes

$$g(\theta) = \langle G_u(\theta), G_v(\theta) \rangle. \tag{4}$$

The Gramians are $k \times k$ PSD matrices, where $k$, the dimension of the embedding space, is much smaller than $n$ – typically $k$ is smaller than 1000, while $n$ can be arbitrarily large. Thus, the Gramian formulation (4) has a much lower computational complexity than the double sum formulation (2), and this reformulation is at the core of alternating least squares and coordinate descent methods (Hu et al., 2008; Bayer et al., 2017), which operate by computing the exact Gramian for one side, and solving for the embeddings on the other. However, these methods do not apply in the non-linear setting due to the implicit dependence on $\theta$, as a change in the model parameters simultaneously changes all embeddings on both sides, making it intractable to recompute the Gramians at each iteration, so the Gramian formulation has not been used when training non-linear models. In the next section, we show that it can in fact be leveraged in the non-linear case.

## 3 TRAINING EMBEDDING MODELS USING GRAMIAN ESTIMATES

In order to leverage the Gramian formulation in the non-linear case, we start by rewriting the objective function (1) in terms of the Gramians defined in (3). Let

$$f_i(\theta) \coloneqq \ell\left(\langle u_i(\theta), v_i(\theta) \rangle, s_i\right) \tag{5}$$

$$g_i(\theta) \coloneqq \frac{1}{2}[\langle u_i(\theta), G_v(\theta) u_i(\theta) \rangle + \langle v_i(\theta), G_u(\theta) v_i(\theta) \rangle], \tag{6}$$

then (1) is equivalent to $\min_{\theta \in \mathbb{R}^d} \frac{1}{n} \sum_{i=1}^{n} [f_i(\theta) + \lambda g_i(\theta)]$. Intuitively, for each example $i$, $-\nabla f_i(\theta)$ pulls the embeddings $u_i$ and $v_i$ close to each other (assuming a high similarity $s_i$), while $-\nabla g_i(\theta)$ creates a repulsive force between $u_i$ and all embeddings $\{v_j\}_{j \in \{1,\dots,n\}}$, and between $v_i$ and all $\{u_j\}_{j \in \{1,\dots,n\}}$, see Appendix D for further discussion, and illustration of the effect of this term. While the Gramians are expensive to recompute at each iteration, we can maintain PSD estimates $\hat{G}_u, \hat{G}_v$ of $G_u(\theta), G_v(\theta)$. Then the gradient of $g(\theta)$ can be approximated by the gradient (w.r.t. $\theta$) of

$$\hat{g}_i(\theta, \hat{G}_u, \hat{G}_v) \coloneqq \left\langle u_i(\theta), \hat{G}_v u_i(\theta) \right\rangle + \left\langle v_i(\theta), \hat{G}_u v_i(\theta) \right\rangle, \tag{7}$$

as stated in the following proposition.

**Proposition 1.** *If $i$ is drawn uniformly in $\{1, \dots, n\}$, and $\hat{G}_u, \hat{G}_v$ are unbiased estimates of $G_u(\theta), G_v(\theta)$ and independent of $i$, then $\nabla_\theta \hat{g}_i(\theta, \hat{G}_u, \hat{G}_v)$ is an unbiased estimate of $\nabla g(\theta)$.*

In a mini-batch setting, one can further average $\hat{g}_i$ over a batch of examples $i \in B$ (which we do in our experiments), but we will omit batches to keep the notation concise. Next, we propose several methods for computing Gramian estimates $\hat{G}_u, \hat{G}_v$, and discuss their tradeoffs. Since each Gramian can be written as a sum of rank-one terms, e.g. $G_u(\theta) = \frac{1}{n} \sum_{i=1}^{n} u_i(\theta) \otimes u_i(\theta)$, a simple unbiased estimate can be obtained by sampling one term (or a batch) from this sum. We improve on this by using different variance reduction methods, which we discuss in the next two sections.

## 3.1 STOCHASTIC AVERAGE GRAMIAN

Our first method is inspired by the stochastic average gradient (SAG) method (Roux et al., 2012; Defazio et al., 2014; Schmidt et al., 2017), which reduces the variance of the gradient estimates by maintaining a cache of individual gradients, and estimating the full gradient using this cache. Since each Gramian is a sum of outer-products (see equation (3)), we can apply the same technique to estimate Gramians. For all $i \in \{1, \ldots, n\}$, let $\hat{u}_i, \hat{v}_i$ be a cache of the left and right embeddings respectively. We will denote by a superscript $(t)$ the value of a variable at iteration $t$. Let $\hat{S}_u^{(t)} = \frac{1}{n} \sum_{i=1}^{n} \hat{u}_i^{(t)} \otimes \hat{u}_i^{(t)}$, which corresponds to the Gramian computed with the current caches. At each iteration $t$, an example $i$ is drawn uniformly at random and the estimate of the Gramian is given by

$$\hat{G}_u^{(t)} = \hat{S}_u^{(t)} + \beta[u_i(\theta^{(t)}) \otimes u_i(\theta^{(t)}) - \hat{u}_i^{(t)} \otimes \hat{u}_i^{(t)}], \tag{8}$$

and similarly for $\hat{G}_v^{(t)}$. This is summarized in Algorithm 1, where the model parameters are updated using SGD (line 10), but this update can be replaced with any first-order method. Here $\beta$ can take one of the following values: $\beta = \frac{1}{n}$, following SAG (Schmidt et al., 2017), or $\beta = 1$, following SAGA (Defazio et al., 2014). The choice of $\beta$ comes with trade-offs that we briefly discuss below. We will denote the cone of positive semi-definite $k \times k$ matrices by $\mathcal{S}_+^k$.

**Proposition 2.** *Suppose $\beta = \frac{1}{n}$ in (8). Then for all $t$, $\hat{G}_u^{(t)}, \hat{G}_v^{(t)} \in \mathcal{S}_+^k$.*

**Proposition 3.** *Suppose $\beta = 1$ in (8). Then for all $t$, $\hat{G}_u^{(t)}$ is an unbiased estimate of $G_u(\theta^{(t)})$.*

While taking $\beta = 1$ gives an unbiased estimate, note that it does not guarantee that the estimates remain in $\mathcal{S}_+^k$. In practice, this can cause numerical issues, but can be avoided by projecting $\hat{G}_u, \hat{G}_v$ on $\mathcal{S}_+^k$, using their eigenvalue decompositions. The per-iteration cost of maintaining the Gramian estimates is $\mathcal{O}(k)$ to update the caches, $\mathcal{O}(k^2)$ to update the estimates $\hat{S}_u, \hat{S}_v, \hat{G}_u, \hat{G}_v$, and $\mathcal{O}(k^3)$ for projecting on $\mathcal{S}_+^k$. Given the small size of the embedding dimension $k$, $\mathcal{O}(k^3)$ remains tractable. The memory cost is $\mathcal{O}(nk)$, since each embedding needs to be cached (plus a negligible $\mathcal{O}(k^2)$ for storing the Gramian estimates). This makes SAGram much less expensive than applying the original SAG(A) methods, which require maintaining caches of the *gradients*, this would incur a $\mathcal{O}(nd)$ memory cost, where $d$ is the number of parameters of the model, and can be orders of magnitude larger than the embedding dimension $k$. However, $\mathcal{O}(nk)$ can still be prohibitively expensive when $n$ is very large. In the next section, we propose a different method which does not incur this additional memory cost.

---

**Algorithm 1** SAGram (Stochastic Average Gramian)

---

1: **Input:** Training data $\{(x_i, y_i, s_i)\}_{i \in \{1, \ldots, n\}}$, learning rate $\eta > 0$.
2: **Initialization phase**
3:     draw $\theta$ randomly
4:     $\hat{u}_i \leftarrow u_i(\theta), \ \hat{v}_i \leftarrow v_i(\theta) \quad \forall i \in \{1, \ldots, n\}$
5:     $\hat{S}_u \leftarrow \frac{1}{n} \sum_{i=1}^{n} \hat{u}_i \otimes \hat{u}_i, \ \hat{S}_v \leftarrow \frac{1}{n} \sum_{i=1}^{n} \hat{v}_i \otimes \hat{v}_i$
6: **repeat**
7:     Update Gramian estimates ($i \sim \text{Uniform}(n)$)
8:         $\hat{G}_u \leftarrow \hat{S}_u + \beta[u_i(\theta) \otimes u_i(\theta) - \hat{u}_i \otimes \hat{u}_i], \quad \hat{G}_v \leftarrow \hat{S}_v + \beta[v_i(\theta) \otimes v_i(\theta) - \hat{v}_i \otimes \hat{v}_i]$
9:     Update model parameters then update caches ($i \sim \text{Uniform}(n)$)
10:         $\theta \leftarrow \theta - \eta \nabla_\theta [f_i(\theta) + \lambda \hat{g}_i(\theta, \hat{G}_u, \hat{G}_v)]$
11:         $\hat{S}_u \leftarrow \hat{S}_u + \frac{1}{n}[u_i(\theta) \otimes u_i(\theta) - \hat{u}_i \otimes \hat{u}_i], \quad \hat{S}_v \leftarrow \hat{S}_v + \frac{1}{n}[v_i(\theta) \otimes v_i(\theta) - \hat{v}_i \otimes \hat{v}_i]$
12:         $\hat{u}_i \leftarrow u_i(\theta), \ \hat{v}_i \leftarrow v_i(\theta)$
13: **until** stopping criterion

---

### 3.2 STOCHASTIC ONLINE GRAMIAN

To derive the second method, we reformulate problem (1) as a two-player game. The first player optimizes over the parameters of the model $\theta$, the second player optimizes over the Gramian estimates $\hat{G}_u, \hat{G}_v \in \mathcal{S}_+^k$, and they seek to minimize the respective losses

$$\begin{cases} L_1^{\hat{G}_u, \hat{G}_v}(\theta) = \frac{1}{n} \sum_{i=1}^n [f_i(\theta) + \lambda \hat{g}_i(\theta, \hat{G}_u, \hat{G}_v)] \\ L_2^\theta(\hat{G}_u, \hat{G}_v) = \frac{1}{2} \|\hat{G}_u - G_u(\theta)\|_F^2 + \frac{1}{2} \|\hat{G}_v - G_v(\theta)\|_F^2, \end{cases} \tag{9}$$

where $\hat{g}_i$ is defined in (7), and $\|\cdot\|_F$ denotes the Frobenius norm. To justify this reformulation, we can characterize its first-order stationary points, as follows.

**Proposition 4.** $(\theta, \hat{G}_u, \hat{G}_v) \in \mathbb{R}^d \times \mathcal{S}_+^k \times \mathcal{S}_+^k$ *is a first-order stationary point for (9) if and only if $\theta$ is a first-order stationary point for problem (1) and $\hat{G}_u = G_u(\theta), \hat{G}_v = G_v(\theta)$.*

Several stochastic first-order dynamics can be applied to problem (9), and Algorithm 2 gives a simple instance where each player implements SGD with a constant learning rate. In this case, the updates of the Gramian estimates (line 7) have a particularly simple form, since $\nabla_{\hat{G}_u} L_2^\theta(\hat{G}_u, \hat{G}_v) = \hat{G}_u - G_u(\theta)$ and can be estimated by $\hat{G}_u - u_i(\theta) \otimes u_i(\theta)$, resulting in the update

$$\hat{G}_u^{(t)} = (1 - \alpha)\hat{G}_u^{(t-1)} + \alpha u_i(\theta^{(t)}) \otimes u_i(\theta^{(t)}), \tag{10}$$

and similarly for $\hat{G}_v$. One advantage of this form is that each update performs a convex combination between the current estimate and a rank-1 PSD matrix, thus guaranteeing that the estimates remain in $\mathcal{S}_+^k$, without the need to project. The per-iteration cost of updating the estimates is $\mathcal{O}(k^2)$, and the memory cost is $\mathcal{O}(k^2)$ for storing the Gramians, which are both negligible.

---

**Algorithm 2** SOGram (Stochastic Online Gramian)

---

1: **Input:** Training data $\{(x_i, y_i, s_i)\}_{i \in \{1, \dots, n\}}$, learning rates $\eta > 0$, $\alpha \in (0, 1)$.
2: **Initialization phase**
3:     draw $\theta$ randomly, $\hat{G}_u, \hat{G}_v \leftarrow 0^{k \times k}$
4: **repeat**
5:     Update Gramian estimates ($i \sim$ Uniform)
6:         $\hat{G}_u \leftarrow (1 - \alpha)\hat{G}_u + \alpha u_i(\theta) \otimes u_i(\theta), \quad \hat{G}_v \leftarrow (1 - \alpha)\hat{G}_v + \alpha v_i(\theta) \otimes v_i(\theta)$
7:     Update model parameters ($i \sim$ Uniform)
8:         $\theta \leftarrow \theta - \eta \nabla_\theta [f_i(\theta) + \lambda \hat{g}_i(\theta, \hat{G}_u, \hat{G}_v)]$
9: **until** stopping criterion

---

The update (10) can also be interpreted as computing an online estimate of the Gramian by averaging rank-1 terms with decaying weights, thus we call the method Stochastic Online Gramian. Indeed, we have by induction on $t$, $\hat{G}_u^{(t)} = \sum_{\tau=1}^t \alpha(1-\alpha)^{t-\tau} u_{i_\tau}(\theta^{(\tau)}) \otimes u_{i_\tau}(\theta^{(\tau)})$. Intuitively, averaging reduces the variance of the estimator but introduces a bias, and the choice of the hyper-parameter $\alpha$ trades-off bias and variance. The next proposition quantifies this tradeoff under mild assumptions.

**Proposition 5.** *Let $\bar{G}_u^{(t)} = \sum_{\tau=1}^t \alpha(1-\alpha)^{t-\tau} G_u(\theta^{(\tau)})$. Suppose that there exist $\sigma, \delta > 0$ such that for all $t$, $\mathbb{E}_i \|u_i(\theta^{(t)}) \otimes u_i(\theta^{(t)}) - G_u(\theta^{(t)})\|_F^2 \leq \sigma^2$ and $\|G_u(\theta^{(t+1)}) - G_u(\theta^{(t)})\|_F \leq \delta$. Then $\forall t$,*

$$\mathbb{E} \|\hat{G}_u^{(t)} - \bar{G}_u^{(t)}\|_F^2 \leq \sigma^2 \frac{\alpha}{2 - \alpha} \tag{11}$$

$$\|\bar{G}_u^{(t)} - G_u^{(t)}\|_F \leq \delta(1/\alpha - 1) + (1 - \alpha)^t \|G_u^{(t)}\|_F. \tag{12}$$

The first assumption simply bounds the variance of single-point estimates, while the second bounds the distance between two consecutive Gramians, a reasonable assumption, since in practice the changes in Gramians vanish as the trajectory $\theta^{(\tau)}$ converges. In the limiting case $\alpha = 1$, $\hat{G}_u^{(t)}$ reduces to a single-point estimate, in which case the bias (12) vanishes and the variance (11) is maximal, while smaller values of $\alpha$ decrease variance and increase bias. This is confirmed in our experiments, as discussed in Section 4.2.

### 3.3 Comparison with existing stochastic methods

We conclude this section by showing that candidate sampling methods (see Chen et al. (2016); Yu et al. (2017) for recent surveys) can be reinterpreted in terms of the Gramian formulation (4). These methods work by approximating the double-sum in (1) using a random sample of pairs. Suppose a batch of pairs $(i, j) \in B \times B'$ is sampled[4], and the double sum is approximated by

$$\tilde{g}(\theta) = \frac{1}{|B||B'|} \sum_{i \in B} \sum_{j \in B'} \mu_i \nu_j \left\langle u_i(\theta), v_j(\theta) \right\rangle^2, \tag{13}$$

where $\mu_i, \nu_j$ are the inverse probabilities of sampling $i, j$ respectively (to guarantee that the estimate is unbiased). Then applying a similar transformation to Section 2.2, one can show that

$$\tilde{g}(\theta) = \left\langle \frac{1}{|B|} \sum_{i \in B} \mu_i u_i(\theta) \otimes u_i(\theta), \frac{1}{|B'|} \sum_{j \in B'} \nu_j v_j(\theta) \otimes v_j(\theta) \right\rangle. \tag{14}$$

which is equivalent to computing two batch-estimates of the Gramians. Implementing existing methods using (14) rather than (13) can decrease their computional complexity in the large batch regime, for the following reason: the double-sum formulation (13) involves a sum of $|B||B'|$ dot products of vectors in $\mathbb{R}^k$, thus computing its gradient costs $\mathcal{O}(k|B||B'|)$. On the other hand, the Gramian formulation (14) is the inner product of two $k \times k$ matrices, each involving a sum over the batch, thus computing its gradient costs $\mathcal{O}(k^2 \max(|B|, |B'|))$, which is cheaper when the batch size is larger than the embedding dimension $k$, a common situation in practice. With this formulation, the advantage of SOGram and SAGram becomes clear, as they use more embeddings to estimate Gramians (by caching or online averaging) than would be possible using candidate sampling.

## 4 Experiments

In this section, we conduct large-scale experiments on the Wikipedia dataset (Wikimedia Foundation). Additional experiments on MovieLens (Harper & Konstan, 2015) are given in Appendix F.

### 4.1 Experimental setup

**Datasets** We consider the problem of learning the intra-site links between Wikipedia pages. Given a pair of pages $(x, y) \in \mathcal{X} \times \mathcal{X}$, the target similarity is 1 if there is a link from $x$ to $y$, and 0 otherwise. Here a page is represented by a feature vector $x = (x_{id}, x_{ngrams}, x_{cats})$, where $x_{id}$ is (a one-hot encoding of) the page URL, $x_{ngrams}$ is a bag-of-words representation of the set of n-grams of the page's title, and $x_{cats}$ is a bag-of-words representation of the categories the page belongs to. Note that the left and right feature spaces coincide in this case, but the target similarity is not necessarily symmetric (the links are directed edges). We carry out experiments on subsets of the Wikipedia graph corresponding to three languages: Simple English, French, and English, denoted respectively by `simple`, `fr`, and `en`. These subgraphs vary in size, and Table 1 shows some basic statistics for each set. Each set is partitioned into training and validation using a (90%, 10%) split.

| language | # pages | # links | # ngrams | # cats |
|----------|---------|---------|----------|--------|
| `simple` | 85K | 4.6M | 8.3K | 6.1K |
| `fr` | 1.8M | 142M | 167.4K | 125.3K |
| `en` | 5.3M | 490M | 501.0K | 403.4K |

Table 1: Corpus sizes for each training set.

**Models** We train non-linear embedding models consisting of a two-tower neural network as in Figure 1, where the left and right embedding functions map, respectively, the source and destination page features. The two embedding networks have the same structure: the input feature embeddings are concatenated then mapped through two hidden layers with ReLU activations. The input embeddings are shared between the two networks, and their dimensions are 50 for `simple`, 100 for `fr`, and 120 for `en`. The sizes of the hidden layers are $[256, 64]$ for `simple` and $[512, 128]$ for `fr` and `en`.

---

[4]The simplest variant uses a uniform distribution, but other methods have been proposed, such adaptive sampling (Bengio & Senecal, 2008; Bai et al., 2017), and importance sampling (Bengio & Senecal, 2003).

**Training methods** The model is trained using a squared error loss, $\ell(s, s') = \frac{1}{2}(s - s')^2$, optimized using SAGram, SOGram, and as baseline, SGD with candidate sampling, using different sampling strategies. The experiments reported in this section use a learning rate $\eta = 0.01$, a penalty coefficient $\lambda = 10$, and batch size 1024. These parameters correspond to the best performance of the baseline methods; we report additional results with different hyper-parameter settings in Appendix E. For SAGram and SOGram, a batch $B$ is used in the Gramian updates (line 8 in Algorithm 1 and line 6 in Algorithm 2, where we use a sum of rank-1 terms over the batch), and another batch $B'$ is used in the model parameter update. For the sampling baselines, the double sum is approximated by all pairs in the cross product $(i, j) \in B \times B'$, and for efficiency, we implement them using the Gramian formulation as discussed in Section 3.3, since we operate in a regime where the batch size is an order of magnitude larger than the embedding dimension $k$. In the first baseline method, `uniform`, items are sampled uniformly from the vocabulary (all pages are sampled with the same probability). The other baseline methods implement importance sampling similarly to Bengio & Senecal (2003); Mikolov et al. (2013): in `linear`, the probability is proportional to the number of occurrences of the page in the training set, and in `sqrt`, the probability is proportional to the square root of the number of occurrences.

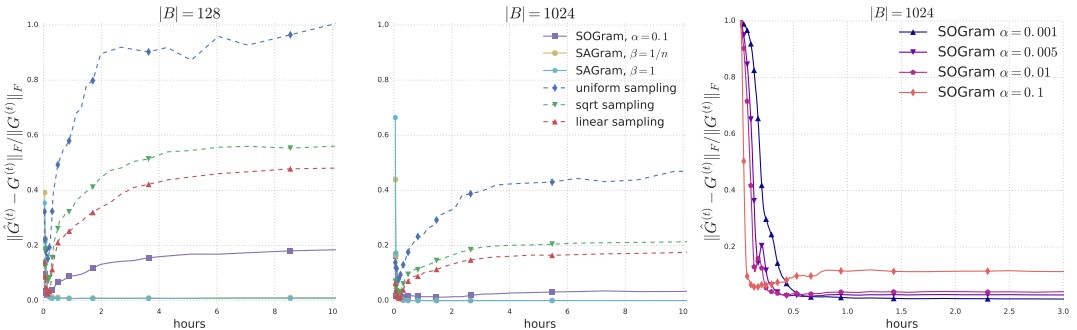

(a) SAGram, SOGram and SGD with different sampling strategies.     (b) SOGram with different averaging rates.

Figure 2: Gramian estimation error on a common trajectory $(\theta^{(t)})$.

## 4.2 QUALITY OF GRAMIAN ESTIMATES

In the first set of experiments, we evaluate the quality of the Gramian estimates using each method. In order to have a meaningful comparison, we fix a trajectory of model parameters $(\theta^{(t)})_{t \in \{1,...,T\}}$, and evaluate how well each method tracks the true Gramians $G_u(\theta^{(t)}), G_v(\theta^{(t)})$ on that common trajectory. This experiment is done on Wikipedia `simple` (the smallest of the datasets) so that we can compute the exact Gramians by periodically computing the embeddings $u_i(\theta^{(t)}), v_i(\theta^{(t)})$ on the full training set at a given time $t$. We report in Figure 2 the estimation error for each method, measured by the normalized Frobenius distance $\frac{\|\hat{G}_u^{(t)} - G_u(\theta^{(t)})\|_F}{\|G_u(\theta^{(t)})\|_F}$. In Figure 2a, we can observe that both variants of SAGram yield the best estimates, and that SOGram yields better estimates than the baselines. Among the baseline methods, importance sampling (both `linear` and `sqrt`) perform better than `uniform`. We also vary the batch size to evaluate its impact: increasing the batch size from 128 to 1024 improves the quality of all estimates, as expected, but it is worth noting that the estimates of SOGram with $|B| = 128$ have comparable quality to baseline estimates with $|B| = 1024$. In Appendix E, we show that a similar effect can be observed for gradient estimates, and we make a formal connection between Gramian and gradient estimation errors.

In Figure 2b, we evaluate the bias-variance tradeoff discussed in Section 3.2, by comparing the estimates of SOGram with different learning rates $\alpha$. We observe that higher values of $\alpha$ suffer from higher variance which persists throughout the trajectory. A lower $\alpha$ reduces the variance but introduces a bias, which is mostly visible during the early iterations.

## 4.3 IMPACT ON TRAINING SPEED AND GENERALIZATION QUALITY

In order to evaluate the impact of the Gramian estimation quality on training speed and generalization, we compare the validation performance of SOGram to the sampling baselines, on each dataset (we

do not use SAGram due to its prohibitive memory cost for corpus sizes of 1M or more). The models are trained with a fixed time budget of 20 hours for `simple`, 30 hours for `fr` and 50 hours for `en`. We estimate the mean average precision (MAP) at 10, by scoring, every 5 minutes, left items in the validation set against 50K random candidates (exhaustively scoring all candidates is prohibitively expensive, but this gives a reasonable approximation). The results are reported in Figure 3. Compared to the sampling baselines, SOGram exhibits faster training and better validation performance across all sampling strategies. Table 2 summarizes the relative improvement of the final validation MAP.

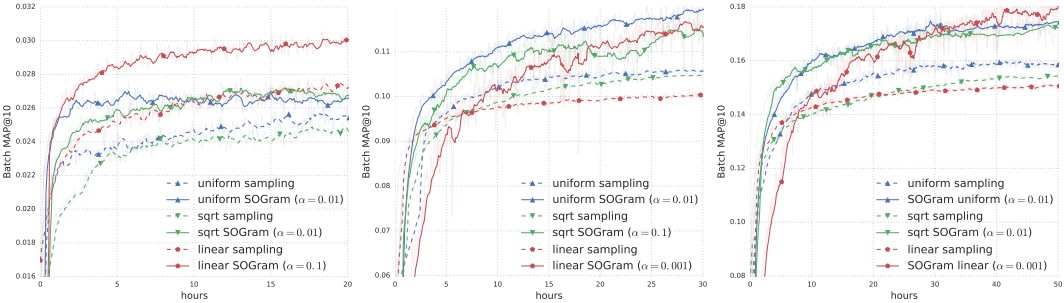

Figure 3: Mean average precision at 10 on the validation set, for different methods, on `simple` (left), `fr` (middle), and `en` (right). The dashed lines correspond to the baseline methods, and the solid lines to SOGram. The different colors represent different sampling strategies.

| language | uniform sampling | uniform SOGram | sqrt sampling | sqrt SOGram | linear sampling | linear SOGram |
|---|---|---|---|---|---|---|
| `simple` | 0.0255 | 0.0266 (+4.2%) | 0.0247 | 0.0268 (+8.3%) | 0.0272 | **0.0300 (+10.7%)** |
| `fr` | 0.1056 | 0.1194 (+13.0%) | 0.1047 | 0.1144 (+9.2%) | 0.1004 | **0.1154 (+15.0%)** |
| `en` | 0.1586 | 0.1743 (+9.9%) | 0.1543 | 0.1723 (+11.7%) | 0.1504 | **0.1797 (19.5%)** |

Table 2: Final validation MAP on each dataset, and relative improvement compared to the baselines.

The improvement on `simple` is modest (between 4% and 10%), which can be explained by the relatively small corpus size (85K unique pages), in which case candidate sampling with a large batch size already yields decent estimates. On the larger corpora, we obtain more significant improvements: between 9% and 15% on `fr` and between 9% and 19% on `en`. It's interesting to observe that the best performance is consistently achieved by SOGram with `linear` importance sampling, even though `linear` performs slightly worse than other strategies in the baseline. SOGram also has a significant impact on training speed: if we measure the time it takes for SOGram to exceed the final validation performance of each baseline method, this time is a small fraction of the total budget. In our experiments, this fraction is between 10% and 17% for `simple`, between 23% and 30% for `fr`, and between 16% and 24% for `en`. Additional numerical results are provided in Appendix E, where we evaluate the impact of other parameters, such as the effect of the batch size $|B|$, the learning rate $\eta$, and the Gramian learning rate $\alpha$. For example, we show that the relative improvement of SOGram compared to the baselines is even larger when using smaller batches, and its generalization performance is more robust to the choice of batch size and learning rate.

## 5 CONCLUSION

We showed that the Gramian formulation commonly used in low-rank matrix factorization can be leveraged for training non-linear embedding models, by maintaining estimates of the Gram matrices and using them to estimate the gradient. By applying variance reduction techniques to the Gramians, one can improve the quality of the gradient estimates, without relying on large sample size as is done in traditional sampling methods. This leads to a significant impact on training time and generalization quality, as indicated by our experiments. While we focused on problems with very large vocabulary size, where traditional approaches are inefficient, it will be interesting to evaluate our methods on other applications such as word-analogy tasks Mikolov et al. (2013); Schnabel et al. (2015). Another direction of future work is to extend this formulation to a larger family of penalty functions, such as the spherical loss family studied in (Vincent et al., 2015; de Brébisson & Vincent, 2016).

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

## A PROOFS

*Proof of Proposition 1.* Starting from the expression (6) of $g(\theta) = \langle G_u(\theta), G_v(\theta) \rangle$, and applying the chain rule, we have

$$\nabla g(\theta) = \nabla \langle G_u(\theta), G_v(\theta) \rangle$$
$$= J_u(\theta)[G_v(\theta)] + J_v(\theta)[G_u(\theta)], \tag{15}$$

where $J_u(\theta)$ denotes the Jacobian of $G_u(\theta)$, an order-three tensor given by

$$J_u(\theta)_{l,i,j} = \frac{\partial G_u(\theta)_{i,j}}{\partial \theta_l}, \qquad l \in \{1, \dots, d\}, i, j \in \{1, \dots, n\},$$

and $J_u(\theta)[G_v(\theta)]$ denotes the vector $[\sum_{i,j} J_u(\theta)_{l,i,j} G_v(\theta)_{i,j}]_{l \in \{1,\dots,d\}}$.

Observing that $\hat{g}_i(\theta, \hat{G}_u, \hat{G}_v) = \langle \hat{G}_u, u_i(\theta) \otimes u_i(\theta) \rangle + \langle \hat{G}_v, v_i(\theta) \otimes v_i(\theta) \rangle$, and applying the chain rule, we have

$$\nabla_\theta \hat{g}_i(\theta, \hat{G}_u, \hat{G}_v) = J_{u,i}(\theta)[\hat{G}_v] + J_{v,i}(\theta)[\hat{G}_u], \tag{16}$$

where $J_{u,i}(\theta)$ is the Jacobian of $u_i(\theta) \otimes u_i(\theta)$, and

$$\mathop{\mathbb{E}}_{i \sim \text{Uniform}}[J_{u,i}(\theta)] = \frac{1}{n} \sum_{i=1}^{n} J_{u,i}(\theta) = J_u(\theta),$$

an similarly for $J_{v,i}$. We conclude by taking expectations in (16) and using assumption that $\hat{G}_u, \hat{G}_v$ are independent of $i$. $\qquad \square$

*Proof of Proposition 2.* From (8) and the definition of $\hat{S}_u^{(t)}$, we have,

$$\hat{G}_u^{(t)} = \frac{1}{n} \sum_{j \neq i} \hat{u}_j^{(t)} \otimes \hat{u}_j^{(t)} + \frac{1}{n} u_i(\theta^{(t)}) \otimes u_i(\theta^{(t)}),$$

which is a sum of matrices in the PSD cone $\mathcal{S}_+^k$. $\qquad \square$

*Proof of Proposition 3.* Denoting by $(\mathcal{F}_t)_{t \geq 0}$ the filtration generated by the sequence $(\theta^{(t)})_{t \geq 0}$, and taking conditional expectations in (8), we have

$$\mathbb{E}[\hat{G}_u^{(t)} | \mathcal{F}_t] = \hat{S}_u^{(t)} + \mathop{\mathbb{E}}_{i \sim \text{Uniform}}[u_i(\theta^{(t)}) \otimes u_i(\theta^{(t)}) - \hat{u}_i^{(t)} \otimes \hat{u}_i^{(t)} | \mathcal{F}_t]$$
$$= \hat{S}_u^{(t)} + \frac{1}{n} \sum_{i=1}^{n}[u_i(\theta^{(t)}) \otimes u_i(\theta^{(t)}) - \hat{u}_i \otimes \hat{u}_i]$$
$$= \frac{1}{n} \sum_{i=1}^{n} u_i(\theta^{(t)}) \otimes u_i(\theta^{(t)}) = G_u(\theta^{(t)}).$$

$\qquad \square$

*Proof of Proposition 4.* $(\theta, \hat{G}_u, \hat{G}_v) \in \mathbb{R}^d \times \mathcal{S}_+^k \times \mathcal{S}_+^k$ is a first-order stationary point of the game if and only if

$$\nabla f(\theta) + \lambda(J_u(\theta)[\hat{G}_v] + J_v(\theta)[\hat{G}_u]) = 0 \tag{17}$$

$$\left\langle \hat{G}_u - G_u(\theta), G' - \hat{G}_u \right\rangle \geq 0, \quad \forall G' \in \mathcal{S}_+^k \tag{18}$$

$$\left\langle \hat{G}_v - G_v(\theta), G' - \hat{G}_v \right\rangle \geq 0, \quad \forall G' \in \mathcal{S}_+^k \tag{19}$$

The second and third conditions simply states that $\nabla_{\hat{G}_u} L_2^\theta(\hat{G}_u, \hat{G}_v)$ and $\nabla_{\hat{G}_v} L_2^\theta(\hat{G}_u, \hat{G}_v)$ define supporting hyperplanes of $\mathcal{S}_+^k$ at $\hat{G}_u, \hat{G}_v$, respectively.

Since $G_u(\theta) \in \mathcal{S}_+^k$, condition (18) is equivalent to $\hat{G}_u = G_u(\theta)$ (and similarly, (19) is equivalent to $\hat{G}_v = G_v(\theta)$). Using the expression (15) of $\nabla g$, we get that (17-19) is equivalent to $\nabla f(\theta) + \lambda \nabla g(\theta) = 0$. $\qquad \square$

*Proof of Proposition 5.* We start by proving the first bound (11). As stated in Section 3.2, we have, by induction on $t$, $\hat{G}_u^{(t)} = \sum_{\tau=1}^{t} a_{t-\tau} u_{i_\tau}(\theta^{(t)}) \otimes u_{i_\tau}(\theta^{(t)})$, where $a_\tau = \alpha(1-\alpha)^\tau$. And by definition of $\bar{G}^{(t)}$, we have $\bar{G}_u^{(t)} = \sum_{\tau=1}^{t} a_{t-\tau} G_u(\theta^{(\tau)})$. Thus,

$$\hat{G}_u^{(t)} - \bar{G}_u^{(t)} = \sum_{\tau=1}^{t} a_{t-\tau} \Delta_u^{(\tau)}$$

where $\Delta_u^{(\tau)} = u_{i_\tau}(\theta^{(\tau)}) \otimes u_{i_\tau}(\theta^{(\tau)}) - G_u(\theta^{(\tau)})$ are zero-mean random variables. Thus, taking the second moment, and using the first assumption (which simply states that the variance of $\Delta_u^{(\tau)}$ is bounded by $\sigma^2$), we have

$$\mathbb{E}\,\|\hat{G}_u^{(t)} - \bar{G}_u^{(t)}\|_F^2 = \mathbb{E}\,\left\|\sum_{\tau=1}^{t} a_{t-\tau} \Delta_u^{(\tau)}\right\|_F^2 = \sum_{\tau=1}^{t} a_{t-\tau}^2\, \mathbb{E}\,\|\Delta_u^{(\tau)}\|_F^2$$

$$\leq \sigma^2 \alpha^2 \sum_{\tau=0}^{t-1} (1-\alpha)^{2\tau} = \sigma^2 \alpha^2 \frac{1-(1-\alpha)^{2t}}{1-(1-\alpha)^2}$$

$$\leq \sigma^2 \frac{\alpha}{2-\alpha},$$

which proves the first inequality (11).

To prove the second inequality, we start from the definition of $\bar{G}_u^{(t)}$:

$$\|\bar{G}_u^{(t)} - G_u^{(t)}\|_F = \|\sum_{\tau=1}^{t} a_{t-\tau}(G_u^{(\tau)} - G_u^{(t)}) - (1-\alpha)^t G_u^{(t)}\|_F$$

$$\leq \sum_{\tau=1}^{t} a_{t-\tau}\|G_u^{(\tau)} - G_u^{(t)}\|_F + (1-\alpha)^t\|G_u^{(t)}\|_F, \tag{20}$$

where the first equality uses that fact that $\sum_{\tau=1}^{t} a_{t-\tau} = 1 - (1-\alpha)^t$. Focusing on the first term, and bounding $\|G_u^{(\tau)} - G_u^{(t)}\|_F \leq (t-\tau)\delta$ by the triangle inequality, we get

$$\sum_{\tau=1}^{t} a_{t-\tau}\|G_u^{(\tau)} - G_u^{(t)})\|_F \leq \delta \sum_{\tau=1}^{t} a_{t-\tau}(t-\tau) = \delta\alpha \sum_{\tau=0}^{t-1} \tau(1-\alpha)^\tau$$

$$= \delta\alpha(1-\alpha)\frac{d}{d\alpha}\left[-\sum_{\tau=0}^{t-1}(1-\alpha)^\tau\right]$$

$$= \delta\alpha(1-\alpha)\frac{d}{d\alpha}\left[-\frac{1-(1-\alpha)^t}{\alpha}\right]$$

$$\leq \delta\alpha(1-\alpha)\frac{1}{\alpha^2}. \tag{21}$$

Combining (20) and (21), we get the desired inequality (12). $\qquad\square$

## B GENERALIZATION TO LOW-RANK PRIORS

So far, we have assumed a uniform zero prior to simplify the notation. In this section, we relax this assumption. Suppose that the prior is given by a low-rank matrix $P = QR^\top$, where $Q, R \in \mathbb{R}^{n \times k_P}$. In other words, the prior for a given pair $(i, j)$ is given by the dot product of two vectors $p_{ij} = \langle q_i, r_j \rangle$. In practice, such a low-rank prior can be obtained, for example, by first training a simple low-rank matrix approximation of the target similarity matrix.

Given this low-rank prior, the penalty term (2) becomes

$$
\begin{aligned}
g^P(\theta) &= \frac{1}{n^2} \sum_{i=1}^{n} \sum_{j=1}^{n} [U_\theta V_\theta^\top - QR^\top]_{ij}^2 \\
&= \frac{1}{n^2} \left\langle U_\theta V_\theta^\top - QR^\top, U_\theta V_\theta^\top - QR^\top \right\rangle \\
&= \frac{1}{n^2} \left[ \left\langle U_\theta^\top U_\theta, V_\theta^\top V_\theta \right\rangle - 2 \left\langle U_\theta^\top Q, V_\theta^\top R \right\rangle + c \right],
\end{aligned}
$$

where $c = \left\langle Q^\top Q, R^\top R \right\rangle$ is a constant that does not depend on $\theta$. Here, we used a superscript $P$ in $g^P$ to disambiguate the zero-prior case.

Now, if we define weighted embedding matrices

$$
\begin{cases}
H_u(\theta) := \frac{1}{n} U_\theta Q = \frac{1}{n} \sum_{i=1}^{n} u_i(\theta) \otimes q_i \\
H_v(\theta) := \frac{1}{n} V_\theta R = \frac{1}{n} \sum_{i=1}^{n} v_i(\theta) \otimes r_i,
\end{cases}
$$

the penalty term becomes

$$
g^P(\theta) = \langle G_u(\theta), G_v(\theta) \rangle - 2 \langle H_u(\theta), H_v(\theta) \rangle + c.
$$

Finally, if we maintain estimates $\hat{H}_u, \hat{H}_v$ of $H_u(\theta), H_v(\theta)$, respectively (using the methods proposed in Section 3), we can approximate $\nabla g^P(\theta)$ by the gradient of

$$
\begin{aligned}
\hat{g}_i^P(\theta, \hat{G}_u, \hat{G}_v, \hat{H}_u, \hat{H}_v) := \\
\left\langle u_i(\theta), \hat{G}_v u_i(\theta) \right\rangle + \left\langle v_i(\theta), \hat{G}_u v_i(\theta) \right\rangle - 2 \left\langle u_i(\theta), \hat{H}_v q_i \right\rangle - 2 \left\langle v_i(\theta), \hat{H}_u r_i \right\rangle. \quad (22)
\end{aligned}
$$

Proposition 1 and Algorithms 1 and 2 can be generalized to the low-rank prior case by adding updates for $\hat{H}_u, \hat{H}_v$, and by using expression (22) of $\hat{g}_i^P$ when computing the gradient estimate.

**Proposition 6.** *If $i$ is drawn uniformly in $\{1, \ldots, n\}$, and $\hat{G}_u$, $\hat{G}_v$, $\hat{H}_u$, $\hat{H}_v$ are unbiased estimates of $G_u(\theta)$, $G_v(\theta)$, $H_u(\theta)$, $H_v(\theta)$, respectively, then $\nabla_\theta \hat{g}_i^P(\theta, \hat{G}_u, \hat{G}_v, \hat{H}_u, \hat{H}_v)$ is an unbiased estimate of $\nabla g^P(\theta)$.*

*Proof.* Similar to the proof of Proposition 1. $\qquad \square$

The generalized versions of SAGram and SOGram are stated below, where the differences compared to the zero prior case are highlighted. Note that, unlike the Gramian matrices, the weighted embedding matrices $H_u, H_v$ are not symmetric, thus we do not project their estimates.

---

**Algorithm 3** SAGram (Stochastic Average Gramian) with low-rank prior

1: **Input:** Training data $\{(x_i, y_i, s_i)\}_{i \in \{1,\ldots,n\}}$, **low-rank priors $\{\mathbf{q_i}, \mathbf{r_i}\}_{\mathbf{i} \in \{1,\ldots,n\}}$**
2: **Initialization phase**
3:    draw $\theta$ randomly
4:    $\hat{u}_i \leftarrow u_i(\theta), \; \hat{v}_i \leftarrow v_i(\theta) \quad \forall i \in \{1, \ldots, n\}$
5:    $\hat{S}_u \leftarrow \frac{1}{n} \sum_{i=1}^{n} \hat{u}_i \otimes \hat{u}_i, \; \hat{S}_v \leftarrow \frac{1}{n} \sum_{i=1}^{n} \hat{v}_i \otimes \hat{v}_i$
6:    $\hat{\mathbf{T}}_{\mathbf{u}} \leftarrow \frac{1}{\mathbf{n}} \sum_{\mathbf{i=1}}^{\mathbf{n}} \hat{\mathbf{u}}_{\mathbf{i}} \otimes \mathbf{q_i}, \quad \hat{\mathbf{T}}_{\mathbf{v}} \leftarrow \frac{1}{\mathbf{n}} \sum_{\mathbf{i=1}}^{\mathbf{n}} \hat{\mathbf{v}}_{\mathbf{i}} \otimes \mathbf{r_i}$
7: **repeat**
8:    Update Gramian estimates ($i \sim \text{Uniform}(n)$)
9:      $\hat{G}_u \leftarrow \hat{S}_u + \beta[u_i(\theta) \otimes u_i(\theta) - \hat{u}_i \otimes \hat{u}_i], \quad \hat{G}_v \leftarrow \hat{S}_v + \beta[v_i(\theta) \otimes v_i(\theta) - \hat{v}_i \otimes \hat{v}_i]$
10:    **Update weighted embedding estimates**
11:      $\hat{\mathbf{H}}_{\mathbf{u}} \leftarrow \hat{\mathbf{T}}_{\mathbf{u}} + \lambda[(\mathbf{u_i}(\theta) - \hat{\mathbf{u}}_{\mathbf{i}}) \otimes \mathbf{q_i}]$
12:      $\hat{\mathbf{H}}_{\mathbf{v}} \leftarrow \hat{\mathbf{T}}_{\mathbf{v}} + \lambda[(\mathbf{v_i}(\theta) - \hat{\mathbf{v}}_{\mathbf{i}}) \otimes \mathbf{r_i}]$
13:    Update model parameters then update caches ($i \sim \text{Uniform}(n)$)
14:      $\theta \leftarrow \theta - \eta \nabla_\theta [f_i(\theta) + \lambda \tilde{p} \hat{g}^P(\theta, \hat{G}_u, \hat{G}_v, \hat{H}_u, \hat{H}_v)]$
15:      $\hat{S}_u \leftarrow \hat{S}_u + \frac{1}{n}[u_i(\theta) \otimes u_i(\theta) - \hat{u}_i \otimes \hat{u}_i], \quad \hat{S}_v \leftarrow \hat{S}_v + \frac{1}{n}[v_i(\theta) \otimes v_i(\theta) - \hat{v}_i \otimes \hat{v}_i]$
16:      $\hat{\mathbf{T}}_{\mathbf{u}} \leftarrow \hat{\mathbf{T}}_{\mathbf{u}} + \frac{1}{\mathbf{n}}[(\mathbf{u_i}(\theta) - \hat{\mathbf{u}}_{\mathbf{i}}) \otimes \mathbf{q_i}], \quad \hat{\mathbf{T}}_{\mathbf{v}} \leftarrow \hat{\mathbf{T}}_{\mathbf{v}} + \frac{1}{\mathbf{n}}[(\mathbf{v_i}(\theta) - \hat{\mathbf{v}}_{\mathbf{i}}) \otimes \mathbf{r_i}]$
17:      $\hat{\mathbf{u}}_{\mathbf{i}} \leftarrow \mathbf{u_i}(\theta), \; \hat{\mathbf{v}}_{\mathbf{i}} \leftarrow \mathbf{v_i}(\theta)$
18: **until** stopping criterion

---

**Algorithm 4** SOGram (Stochastic Online Gramian) with low-rank prior

1: **Input:** Training data $\{(x_i, y_i, s_i)\}_{i \in \{1,\ldots,n\}}$, **low-rank priors $\{\mathbf{q_i}, \mathbf{r_i}\}_{\mathbf{i} \in \{1,\ldots,n\}}$**
2: **Initialization phase**
3:    draw $\theta$ randomly
4:    $\hat{G}_u, \hat{G}_v \leftarrow 0^{k \times k}$
5: **repeat**
6:    Update Gramian estimates ($i \sim \text{Uniform}(n)$)
7:      $\hat{G}_u \leftarrow (1 - \alpha)\hat{G}_u + \alpha u_i(\theta) \otimes u_i(\theta), \quad \hat{G}_v \leftarrow (1 - \alpha)\hat{G}_v + \alpha v_i(\theta) \otimes v_i(\theta)$
8:    **Update weighted embedding estimates**
9:      $\hat{\mathbf{H}}_{\mathbf{u}} \leftarrow (\mathbf{1} - \alpha)\hat{\mathbf{H}}_{\mathbf{u}} + \alpha \mathbf{u_i}(\theta) \otimes \mathbf{q_i}, \quad \hat{\mathbf{H}}_{\mathbf{v}} \leftarrow (\mathbf{1} - \alpha)\hat{\mathbf{H}}_{\mathbf{v}} + \alpha \mathbf{v_i}(\theta) \otimes \mathbf{r_i}$
10:    Update model parameters ($i \sim \text{Uniform}(n)$)
11:      $\theta \leftarrow \theta - \eta \nabla_\theta [f_i(\theta) + \lambda \tilde{p} \hat{g}^P(\theta, \hat{G}_u, \hat{G}_v, \hat{H}_u, \hat{H}_v)]$
12: **until** stopping criterion

---

## C   NON-UNIFORM WEIGHTS

In additional to using a non-uniform prior, it can also be desirable to use non-uniform weights in the penalty term, for example to balance the contribution of frequent and infrequent items to the penalty term. We discuss how to adapt our algorithms to the non-uniform weights case. Suppose that the penalty function is given by

$$g^W(\theta) = \frac{1}{n^2} \sum_{i=1}^{n} \sum_{j=1}^{n} a_i b_j \langle u_i(\theta), v_j(\theta) \rangle^2 ,$$

where $a_i, b_j$ are positive left and right weights, respectively. Here we used a superscript $W$ in $g^W$ to disambiguate the uniform-weight case. Then using a similar transformation to Section 2.2, we can rewrite $g^W$ as follows:

$$g^W(\theta) = \left\langle \frac{1}{n} \sum_{i=1}^{n} a_i u_i \otimes u_i, \frac{1}{n} \sum_{j=1}^{n} b_j v_j \otimes v_j \right\rangle ,$$

i.e. $g^W$ is the inner-product of two weighted Gramians. Both SAGram and SOGram can be generalized to this case, by maintaining estimates of the weighted Gramians, one simply needs to scale the contribution of each term $u_i \otimes u_i$ by the appropriate embedding weight $a_i$ (and similarly for the right embedding).

**Remark** Here we discussed the case of a rank-one weight matrix, i.e. when the unobserved weight matrix can be written as $W = a \otimes b$ for a given left and right weight vectors $a, b$. The weight matrix cannot be arbitrary (as specifying $n^2$ individual weights is prohibitively expensive in many applications such as the experiments of this paper), thus one needs a consice description of the weights matrix. One such description is the sum of a sparse and low-rank matrix, and one can generalize SAGram and SOGram to this case: the sparse part of the weight matrix can be optimized explicitly, and the low-rank part can be optimized using weighted Gramians, by generalizing the argument of the previous paragraph.

## D    INTERPRETATIONS OF THE GRAMIAN PENALTY TERM

In this section, we briefly discuss different interpretations of the Gramian inner-product $g(\theta)$. Starting from the expression (4) of $g(\theta)$ and the definition (3) of the Gram matrices, we have

$$g(\theta) = \langle G_u(\theta), G_v(\theta) \rangle = \left\langle \frac{1}{n} \sum_{i=1}^{n} u_i(\theta) \otimes u_i(\theta), G_v(\theta) \right\rangle = \frac{1}{n} \sum_{i=1}^{n} \langle u_i(\theta), G_v(\theta) u_i(\theta) \rangle, \quad (23)$$

which is a quadratic form in the left embeddings $u_i$ (and similarly for $v_j$, by symmetry). In particular, the partial derivative of the Gramian term with respect to an embedding $u_i$ is

$$\frac{\partial g(\theta)}{\partial u_i} = \frac{2}{n} G_v(\theta) u_i(\theta) = \frac{2}{n} \left[ \frac{1}{n} \sum_{j=1}^{n} v_j(\theta) \otimes v_j(\theta) \right] u_i(\theta).$$

Each term $(v_j \otimes v_j)u_i = v_j \langle v_j, u_i \rangle$ is simply the projection of $u_i$ on $v_j$ (scaled by $\|v_j\|^2$). Thus the gradient of $g(\theta)$ with respect to $u_i$ is an average of scaled projections of $u_i$ on each of the right embeddings $v_j$, and moving in the direction of the negative gradient simply moves $u_i$ away from regions of the embedding space with a high density of right embeddings. This corresponds to the intuition discussed in the introduction: the purpose of the $g(\theta)$ term is precisely to push left and right embeddings away from each other, to avoid placing embeddings of dissimilar items near each other, a phenomenon referred to as folding of the embedding space (Xin et al., 2017).

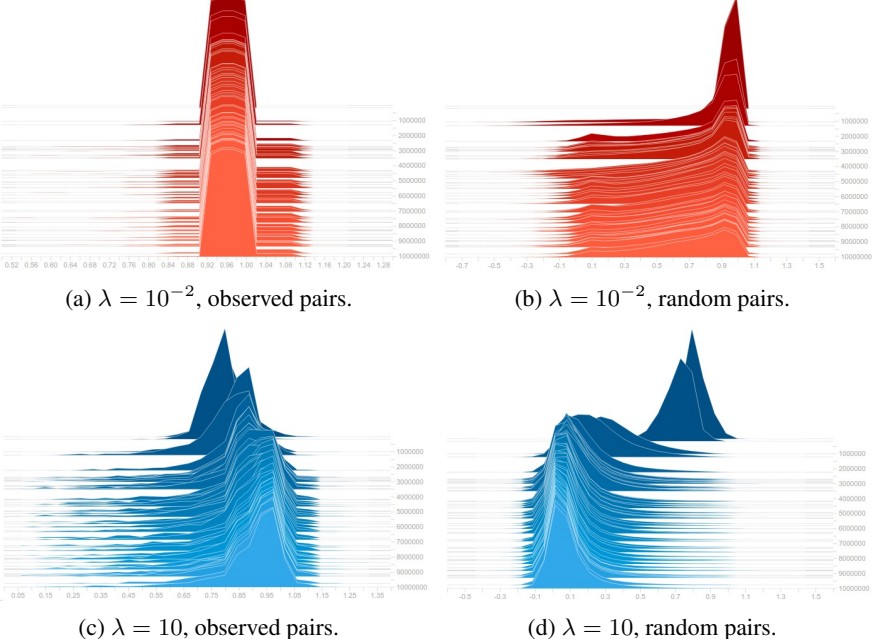

(a) $\lambda = 10^{-2}$, observed pairs.    (b) $\lambda = 10^{-2}$, random pairs.

(c) $\lambda = 10$, observed pairs.    (d) $\lambda = 10$, random pairs.

Figure 4: Evolution of the inner product distribution $\langle u_i(\theta^{(t)}), v_j(\theta^{(t)}) \rangle$ in the Wikipedia `en` model trained with different penalty coefficients $\lambda$, for observed pairs (left) and random pairs (right).

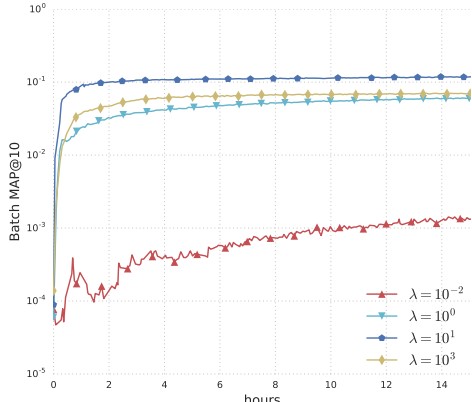

Figure 5: Mean Average Precision of the Wikipedia `en` model, trained with different values of the penalty coefficient $\lambda$.

In order to illustrate the effect of this term on the embedding distributions, we visualize, in Figure 4, the distribution of the inner product $\langle u_i(\theta^{(t)}), v_j(\theta^{(t)}) \rangle$, for random pairs $(i, j)$, and for observed pairs $(i = j)$, and how these distributions change as $t$ increases. The plots are generated for the Wikipedia `en` model described in Section 4, trained with SOGram ($\alpha = 0.01$), with two different values of the penalty coefficient, $\lambda = 10^{-2}$ and $\lambda = 10$. In both cases, the distribution for observed pairs remains concentrated around values close to 1, as one expects (recall that the target similarity is 1 for observed pairs, i.e. pairs of connected pages in the Wikipedia graph). The distributions for random pairs, however, are very different: with $\lambda = 10$, the distribution quickly concentrates around a value close to 0, while with $\lambda = 10^{-2}$ the distribution is more flat, and a large proportion of pairs have a high inner-product. This indicates that with a lower $\lambda$, the model is more likely to fold, i.e. place embeddings of unrelated items near each other. This is consistent with the validation MAP, reported in Figure 5. With $\lambda = 10^{-2}$, the validation MAP increases very slowly, and remains two orders of magnitude smaller than the model trained with $\lambda = 10$. The figure also shows that when $\lambda$ is too large ($\lambda = 10^3$), the model is over-regularized and the MAP decreases.

To conclude this section, we note that our methods also apply to a related regularizer introduced in (Zhang et al., 2017), called Global Orthogonal Regularization. The authors argue that when learning feature embedding representations, spreading out the embeddings is helpful for generalization, and propose to match the second moment of each embedding distribution with that of the uniform distribution. Formally, and using our notation, they use the penalty term $\max(g_u(\theta), 1/k) + \max(g_v(\theta), 1/k)$, where $k$ is the embedding dimension, $g_u(\theta) = \frac{1}{n^2} \sum_{i=1}^{n} \sum_{j=1}^{n} \langle u_i, u_j \rangle^2$, and similarly for $g_v$. They optimize this term using candidate sampling. We can also apply the same Gramian transformation as in Section 2.2 to write $g_u(\theta) = \langle G_u(\theta), G_u(\theta) \rangle$, and $g_v(\theta) = \langle G_v(\theta), G_v(\theta) \rangle$, and we can similarly apply SAGram and SOGram to estimate both Gramians. Formally, the difference here is that one would penalize the inner-product of each Gramian with itself, instead of the inner-product of the two. One advantage of this regularizer is that it applies to a broader class of models, as it does not require the output of the model to be the dot-product of two embedding functions.

## E  FURTHER EXPERIMENTS ON WIKIPEDIA

### E.1  QUALITY OF GRADIENT ESTIMATES

The experiments in Section 4 indicate that our methods give better estimates of the Gramians, and a natural question is how this affects gradient estimation quality. First, one can make a formal connection between the two. Since

$$\frac{\partial g(\theta)}{\partial u_i} = \frac{2}{n} G^V(\theta) u_i(\theta)$$

$$\frac{\partial \hat{g}(\theta)}{\partial u_i} = \frac{2}{n} \hat{G}^V u_i(\theta),$$

the estimation error of the gradient with respect to the left embeddings $u$ is

$$\|\nabla_u g - \nabla_u \hat{g}\|_2^2 = \frac{4}{n^2} \sum_{i=1}^n \|(G_v - \hat{G}_v)u_i\|^2$$

$$= \frac{4}{n^2} \sum_{i=1}^n \left\langle (G_v - \hat{G}_v)u_i, (G_v - \hat{G}_v)u_i \right\rangle$$

$$= \frac{4}{n^2} \sum_{i=1}^n \left\langle G_v - \hat{G}_v, (G_v - \hat{G}^V)u_i \otimes u_i \right\rangle$$

$$= \frac{4}{n} \left\langle G_v - \hat{G}_v, (G_v - \hat{G}_v)G_u \right\rangle.$$

This last expression can be interpreted as a Frobenius norm of the right Gramian estimation error $G_v - \hat{G}_v$, weighted by the left Gramian $G_u$, thus the gradient error is closely related to the Gramian error. Figure 6 shows the gradient estimation quality on Wikipedia `simple`, measured by the normalized squared norm $\frac{\|\nabla_u \hat{g} - \nabla_u g\|_2}{\|\nabla_u g\|_2}$. The results are similar to the Gramian estimation errors reported in Figure 2.

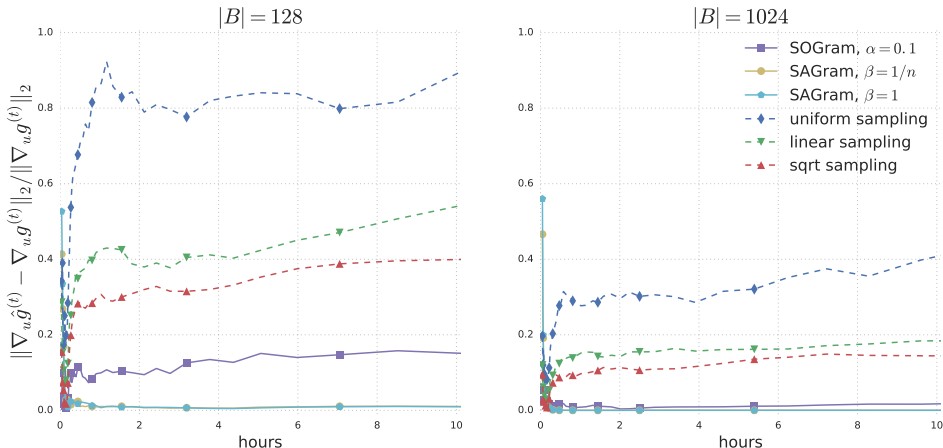

Figure 6: Gradient estimation error on a common trajectory $\theta^{(t)}$, with batch sizes $|B| = 128$ (left) and $|B| = 1024$ (right).

### E.2 EFFECT OF THE SAMPLING DISTRIBUTION

Comparing the different baselines methods on `simple` (Figure 3), we observe that `uniform` sampling performs better than `sqrt`, despite having a worse Gramian estimate according to Figure 2a. One possible explanation is that the sampling distribution affects both the quality of the Gramian estimates, and the frequency at which the item embeddings are updated, which in turn affects the MAP. In particular, tail items are updated more frequently under `uniform` than other distributions, and this may have a positive impact on the MAP.

### E.3 EFFECT OF GRAMIAN LEARNING RATE $\alpha$ AND BIAS-VARIANCE TRADEOFF

In addition to the experiments of Section 4, we also evaluated the effect of the Gramian learning rate $\alpha$ on the quality of the Gramian esimates and generalization performance on Wikipedia `en`. Figure 7 shows the validation MAP of the SOGram method for different values of $\alpha$ (together with the basline for reference). This reflects the bias-variance tradeoff dicussed in Proposition 5: with a lower $\alpha$, progress is initially slower (due to the bias introduced in the Gramian estimates), but the final performance is better. Given a limited training time budget, this suggests that a higher $\alpha$ can be preferable.

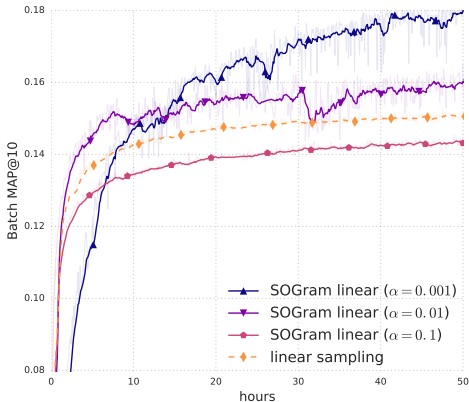

Figure 7: Validation MAP of SOGram and linear importance sampling, on Wikipedia `en`, for different values of the Gramian learning rates $\alpha$.

We also evaluate the quality of the Gramian estimates, but due to the large vocabulary size in `en`, computing the exact Gramians is no longer feasible, so we approximate it using a large sample of 1M embeddings. The results are reported in Figure 8, which shows the normalized Frobenius distance between the Gramian estimates $\hat{G}_u$ and (the large sample approximation of) the true Gramian $G_u$. The results are similar to the experiment on `simple`: with a lower $\alpha$, the estimation error is initially high, but decays to a lower value as training progresses, which can be explained by the bias-variance tradeoff discussed in Proposition 5.

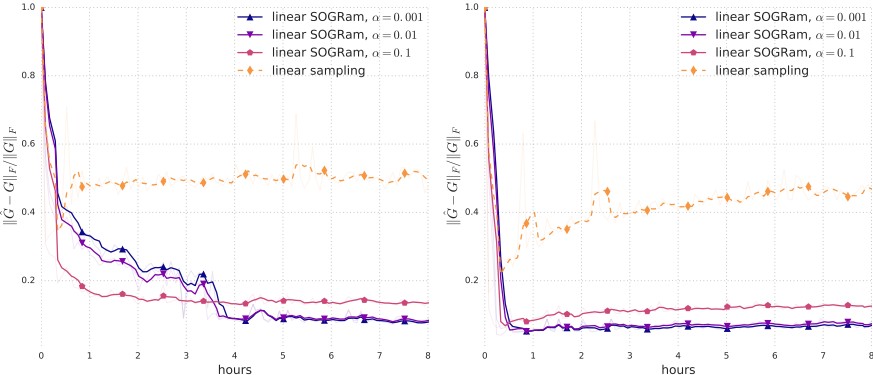

Figure 8: Gramian estimation error on `en`, for SOGram with different values of $\alpha$, and different learning rates. The left and right figures correspond respectively to $\eta = 0.01$ and $\eta = 0.002$.

The tradeoff is affected by the trajectory of the true Gramians: smaller changes in the Gramians (captured by the parameter $\delta$ in Proposition 5) induce a smaller bias. In particular, changing the learning rate $\eta$ of the main algorithm can affect the performance of the Gramian estimates by affecting the rate of change of the true Gramians. To investiage this effect, we ran the same experiment with two different learning rates, $\eta = 0.01$ as in Section 4, and a lower learning rate $\eta = 0.002$. The errors converge to similar values in both cases, but the error decay occurs much faster with smaller $\eta$, which is consistent with our analysis.

### E.4 EFFECT OF BATCH SIZE AND LEARNING RATE

In this section, we explore the effect of the batch size $|B|$ and learning rate $\eta$ on the performance of SOGram compared to the baselines. We ran the Wikipedia `en` experiment with different values of these hyperparameters, and report the final validation MAP in Tables 3 and 4, which correspond to batch size $128$ and $512$ respectively.

|  | linear sampling | linear SOGram ($\alpha = 0.1$) | linear SOGram ($\alpha = 0.01$) | linear SOGram ($\alpha = 0.001$) |
|---|---|---|---|---|
| $\eta = 0.02$ | 0.0896 | 0.1249 (+39.5%) | 0.1256 (+40.2%) | 0.1683 (+87.9%) |
| $\eta = 0.01$ | 0.1325 | 0.1286 (-2.9%) | 0.1301 (-1.8%) | **0.1710 (+29.1%)** |
| $\eta = 0.005$ | 0.1290 | 0.1300 (+0.8%) | 0.1270 (-1.5%) | 0.1385 (+7.4%) |

Table 3: Final validation MAP on Wikipedia `en`, with batch size $|B| = 128$.

|  | linear sampling | linear SOGram ($\alpha = 0.1$) | linear SOGram ($\alpha = 0.01$) | linear SOGram ($\alpha = 0.001$) |
|---|---|---|---|---|
| $\eta = 0.02$ | 0.0519 | 0.1331 (+156.4%) | 0.1198 (+130.7%) | 0.1295 (+149.3%) |
| $\eta = 0.01$ | 0.1371 | 0.1329 (-3.0%) | 0.1505 (+9.8%) | **0.1737 (+26.7%)** |
| $\eta = 0.005$ | 0.1346 | 0.1323 (-1.7%) | 0.1299 (-3.5%) | 0.1569 (+16.6%) |

Table 4: Final validation MAP on Wikipedia `en`, with batch size $|B| = 512$.

We can make several observations. First, the best performance is consistently achieved by SOGram with learning rate $\alpha = 0.001$. Second, the relative improvement compared to the baseline is, in general, larger for smaller batch sizes. This can be explained intuitively by the fact that because of online averaging, the quality of the Gramian estimates with SOGram suffers less than with the sampling baseline. Finally, we can also observe that the final performance also seems more robust to the choice of batch size and learning rate, compared to the baseline. For example, with the larger learning rate $\eta = 0.02$, the performance degrades for all methods, but the drop in performance for the baseline is much more significant than for the SOGram methods.

## F    EXPERIMENT ON MOVIELENS DATA

In this section, we report experiments on a regression task on MovieLens.

**Dataset**    The MovieLens dataset consists of movie ratings given by a set of users. In our notation, the left features $x$ represent a user, the right features $y$ represent an item, and the target similarity is the rating of movie $y$ by user $x$. The data is partitioned into a training and a validation set using a (80%-20%) split. Table 5 gives a basic description of the data size. Note that it is comparable to the `simple` dataset in the Wikipedia experiments.

| Dataset | # users | # movies | # ratings |
|---|---|---|---|
| MovieLens | 72K | 10K | 10M |

Table 5: Corpus size of the MovieLens dataset.

**Model**    We train a two-tower neural network model, as described in Figure 1, where each tower consists of an input layer, a hidden layer, and output embedding dimension $k = 35$. The left tower takes as input a one-hot encoding of a unique user id, and the right tower takes as input one-hot encodings of a unique movie id, the release year of the movie, and a bag-of-words representation of the genres of the movie. These input embeddings are concatenated and used as input to the right tower.

**Methods**    The model is trained using a squared loss $\ell(s, s') = \frac{1}{2}(s - s')^2$, using SOGram with different values of $\alpha$, and sampling as a baseline. We use a learning rate $\eta = 0.05$, and penalty coefficient $\lambda = 1$. We measure mean average precision on the trainig set and validation set, following the same procedure described in Section 4. The results are given in Figure 9.

**Results**    The results are similar to those reported on the Wikipedia `simple` dataset, which is comparable in corpus size and number of observations to MovieLens. The best validation mean average precision is achieved by SOGram with $\alpha = 0.1$ (for an improvement of 2.9% compared to the sampling baseline), despite its poor performance on the training set, which indicates that better

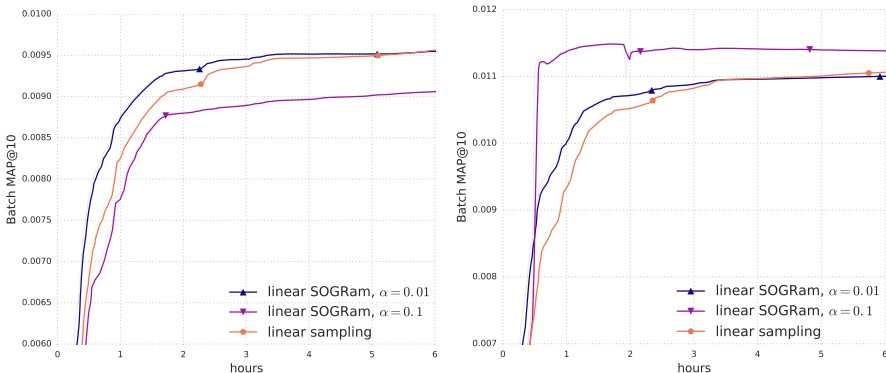

Figure 9: Mean average precision at 10 on the training set (left) and the validation set (right), for different methods, on the MovieLens dataset.

estimation of $g(\theta)$ induces better regularization. The impact on training speed is also remarkable in this case, SOGram with $\alpha = 0.1$ achieves a better validation performance in under 1 hour of training than the sampling baseline in 6 hours.

