# OpenReview forum: "Efficient Training on Very Large Corpora via Gramian Estimation"
_ICLR.cc/2019/Conference_

### Official Review · AnonReviewer1 · 2018-10-31
**Good work overall**

**Rating:** 7
**Confidence:** 2

**Review:**

This paper proposes a method for estimating non-linear similarities between items using Gramian estimation. This is achieved by having two separate neural networks defined for each item to be compared, which are then combined via a dot product. The proposed innovation in this paper is to use Gramian estimation for the penalty parameter of the optimization which allows for the non-linear case. Two algorithms are proposed which allow for estimation in the stochastic / online setting. Experiments are presented which appear to show good performance on some standard benchmark tasks.

Overall, I think this is an interesting set of ideas for an important problem. I have two reservations. First, the organization of the paper needs to be addressed in order to aid user readability. The paper often jumps across sections without giving motivation or connecting language. This will limit the audience of the paper and the work. Second (and more importantly), I found the experiments to be slightly underwhelming. The hyperparameters (batch size, learning rate) and architecture don’t have any rationale attached to them. It is also not entirely clear whether the chosen comparison methods fully constitute the current state of the art. Nonetheless, I think this is an interesting idea and strong work with compelling results.

Editorial comments:

The organization of this paper leaves something to be desired. The introductions ends very abruptly, and then appears to begin again after the related work section. From what I can tell the first three sections all constitute the introduction and should be merged with appropriate edits to make the narrative clear.

“where x and y are nodes in a graph and the similarity is wheter an edge” → typo and sentence ends prematurely.

---

> ### Author Response · Authors · 2018-11-16
> **Thank you for your review**
>
> Thank you for your review and your helpful suggestions.
>
> We updated the organization following the reviewer's suggestions, by reorganizing the introduction and improving the transitions between sections. We also added a comment about our choice of hyper-parameters: in the main experiments of Section 4, the hyper-parameters were cross-validated using the baseline. The effect of some of the hyper-parameters is further studied in the appendix: the effect of the batch size and learning rate is studied in Appendix D.2 (now Appendix E.4 in the revision), and the effect of the penalty coefficient λ is illustrated in Appendix C (now Appendix D in the revision). We did not include these results in the main body of the paper for space constraints, and to keep the message focused, but we added a note to Section 4 pointing to the appendix for further details on the effect of the various hyper-parameters.

---

### Official Review · AnonReviewer2 · 2018-11-02
**Nice work**

**Rating:** 8
**Confidence:** 4

**Review:**

This paper proposes an efficient algorithm to learn  neural embedding models with a dot-product structure over very large corpora. The main method is to reformulate the objective function in terms of generalized Gramiam matrices, and maintain estimates of those matrices in the training process. The algorithm uses less time and achieves significantly better quality than sampling based methods.

1. About the experiments, it seems the sample size for sampling based experiments is not discussed. The number of noise samples have a large influence on the performance of the models. In figure 2, different sampling strategies are discussed. It would be cool if we can also see how the sampling size affects the estimation error.

2. If we just look at the sampling based methods, in figure 2a, uniform sampling’s Gramian estimates is the worst. But the MAP of uniform sampling on validation set for all three datasets are not the worst. Do you have any comments?

3. wheter an edge -> whether an edge.

---

> ### Author Response · Authors · 2018-11-16
> **Thank you for your review**
>
> Thank you for your review and your helpful suggestions.
>
> 1) On the effect of sample size: we agree that the sample size directly affects the performance of these methods. We investigated this effect in Appendix D.2 (which is now Appendix E.4 in the revision), where we ran the same experiment on Wikipedia English with batch sizes 128, 512 (Tables 3 and 4), and compared the results to batch size 1024 (Table 2). We simultaneously varied the learning rate to understand its effect as well, but focusing on the effect of batch size only, we can observe that
> (i) the performance of all methods increases with the batch size (at least in the 128-1024 range).
> (ii) the relative improvement of our methods (compared to the baseline) is larger for smaller batch sizes: the relative improvement is 19.5% for 1024, 26.7% for 512, and 29.1% for 128.
> Of course, one cannot increase the batch size indefinitely as there are hard limits on memory size, and the key advantage of our methods is in problems where sampling-based methods give poor estimates even with the largest feasible batch size.
> The effect of the batch size can also be seen to some extent in Figure 2.a, where we show the quality of the Gramian estimates for batch size 128 and 1024. The figure suggests that the quality improves, for all methods, with larger batch sizes, and that SOGram with batch size 128 has a comparable estimation quality to the baseline with batch size 1024.
>
> 2) The reviewer raises an interesting point. We have observed in our experiments that for a fixed sampling distribution, improving the Gramian estimates generally leads to better MAP, but we cannot draw conclusions when the sampling distribution changes. One possible explanation is that the sampling distribution affects both the quality of the Gramian estimates, and the frequency at which the item embeddings are updated. In particular, tail items are sampled more often under uniform sampling than under the other distributions, and updating their embeddings more frequently may contribute to improving the MAP. We added a comment (Appendix E.2 in the revision) to highlight this observation.

---

### Official Review · AnonReviewer3 · 2018-11-03
**Good paper with clear contribution, could be made stronger with better evaluation**

**Rating:** 7
**Confidence:** 4

**Review:**

Summary of the paper:

This work presents a novel method for similarity function learning using non-linear model. The main problem with the similarity function learning models is the pairwise component of the loss function which grows quadratically with the training set. The existing stochastic approximations which are agnostic to training set size have high variance and this in-turn results in poor convergence and generalisation. This paper presents a new stochastic approximation of the pairwise loss with reduced variance. This is achieved by exploiting the dot-product structure of the least-squares loss and is computationally efficient provided the embedding dimensions are small. The core idea is to rewrite the least-squares as the matrix dot product of two PSD matrices (Grammian). The Grammian matrix is the sum of the outer-product of embeddings along the training samples. The authors present two algorithms for training the model, 1)SAGram: By maintaining a cache of all embedding vectors of training points (O(nk) space)$, whenever a point is encountered it's cache is replaced with it's embedding vector. 2) SOGram: This algorithm keeps a moving average of the Grammian estimate to reduce the variance. Experimental results shows that this approach reduces the variance in the Grammian estimates, results in faster convergence and better generalisation.

Review:

The paper is well written with clear contribution to the problem of similarity  learning.  My only complain is that, I think the evaluation is a bit weak and does not support the claim that is applicable all kinds of problems e.g. nlp and recommender systems. This task in Wikipedia does not seem to be standard (kind of arbitrary) — there are some recommendation results in the appendix but I think it should have been in the main paper.

Overall interesting but I would recommend evaluating in standard similarity learning for nlp and other tasks (perhaps more than one)

There are specific similarity evaluation sets for word embeddings. It can be found in following papers: https://arxiv.org/pdf/1301.3781.pdf
http://www.aclweb.org/anthology/D15-1036

---

> ### Author Response · Authors · 2018-11-16
> **Thank you for your review**
>
> Thank you for your assessment and your helpful suggestions.
>
> Regarding evaluation: since the focus of the paper is on the design of an efficient optimization method, we wanted to choose an experiment where (i) the evaluation metric is aligned with the optimization objective, and (ii) the vocabulary size is very large (on the order of 10^6 or more), making traditional sampling-based methods inefficient, because they would require too many samples to achieve high model quality. This is why we chose the Wikipedia dataset, which is, to our knowledge, one of the few publicly available datasets of this scale. It also offers different subsets of varying scale, which allowed us to illustrate the effect of the problem size, suggesting that the benefit of the Gramian-based methods increases with vocabulary size. We added a note to the revision to comment on our choice.
> We also agree that it will be beneficial to evaluate these method on other applications such as more traditional natural language tasks, and this is something we intend to pursue in future work.

---

### Public Comment · ~Yu_Bai1 · 2018-10-05
**Interesting idea**

This paper studies the problem of learning embeddings on large corpora, and proposes to replace the commonly used sampling mechanism by an online Gramian estimate. It seems like the proposed Gramian estimate allows a lot of information reuse (which is otherwise lost in baseline sampling methods) and hence improves the training.

I liked the idea of maintaining an estimate of an important (and relatively small-sized) quantity to allow information reuse, and I think it has the potential to be generalized into similar types of problems as well.

A question about the experiment: in Section 4.2 it is shown that the maintained Gramian estimates are indeed better than sampling estimates. Perhaps a similar test can be done on the gradients, and hopefully the stochastic gradients given by the Gramian estimate are indeed closer to the full gradient, compared with the baseline sampling methods?

---

> ### Author Response · Authors · 2018-10-14
> **Quality of gradient estimates**
>
> Thank you for your comments and for the suggestion.
> First, one can make a formal connection between the quality of Gramian estimates and the quality of gradient estimates.
> The prior term can be written as  1/𝑛 ∑_i ⟨𝑢_𝑖, 𝐺^𝑣 𝑢_𝑖⟩ , thus the partial derivative w.r.t.  𝑢_𝑖  is  ∇_𝑢𝑖 𝑔 = 2/𝑛 𝐺^𝑣 𝑢_𝑖 . If the Gramian  𝐺^𝑣  is approximated by Ĝ^𝑣 , then the gradient estimation error is  2/𝑛 ∑_i ‖(𝐺^𝑣− Ĝ^𝑣) 𝑢_𝑖‖^2 = 2/𝑛 ∑_i ⟨(𝐺^𝑣 − Ĝ^𝑣)𝑢_𝑖,(𝐺^𝑣 − Ĝ^𝑣)𝑢_𝑖⟩  which is equal to 2⟨(𝐺^𝑣 − Ĝ^𝑣),(𝐺^𝑣 − Ĝ^𝑣)𝐺^𝑢⟩ , in other words, the estimation error of the right gradient is the "𝐺^𝑢 -weighted" Frobenius norm of the left Gramian error.
> We generated these plots as suggested, on Wikipedia simple, and we observe the same trend for the gradient estimation errors as the Gramian estimation error in Figure 2.a. We will include this experiment in the updated version of the paper during rebuttal. Thanks again for the suggestion.

---

### Public Comment · (anonymous) · 2018-10-08
**Very good papers.**

Definitely, it's a good paper.
Sampling-based methods has dominated the main trend for many years, through BPR in recommendation field and negative sampling in word embedding. Some previous research proposed to train from whole data while their methods only focused on  shadow linear models like matrix factorization. This paper proposed to extend the framework of learning from whole data to deep learning based embeddings by using Gramian estimates.
Several questions:
1. Although the proposed scheme can get rid of sampling, the final layer must be an inner product. Will it limit the performance of the model?
2.The hyperparameter lambda is defined as the weight for negative samples. Is it reasonable to assign a uniform weight for all samples?
3.Could you please public the code for one of your evaluation tasks?

---

> ### Author Response · Authors · 2018-10-14
> **Non-uniform weights**
>
> Thank you for your comments, we will discuss each point below.
>
> 1) The dot-product structure is important in many applications, especially in retrieval with very large corpora (since it allows efficient scoring using maximum-inner product search techniques [1, 2]). In addition to dot-product models, our methods can also be useful in more general architectures when used jointly with the Global Orthogonal regularizer proposed in [3], which "spreads-out" the embeddings by pushing the embedding distribution towards the uniform distribution. This was shown to improve generalization performance. In the last paragraph of Appendix C, we show that the Global Orthogonal regularizer can be written in terms of Gramians, thus our methods can be used in such models.
>
> 2) Using non-uniform weights can be important, and it is supported by the methods we propose. They also support the use of a non-uniform sampling distribution, and non-uniform prior (as discussed in Appendix B). For non-uniform weights, if we define the weight of a left item i to be  𝑎_𝑖  and the weight of a right item  𝑗  to be  𝑏_𝑗 , and define the penalty term as  1/𝑛^2 ∑_𝑖 ∑_𝑗 𝑎_𝑖 𝑏_𝑗 ⟨𝑢_𝑖, 𝑣_𝑗⟩^2, then one can show, using the same argument as in Section 2.2, that this is equal to the matrix inner-product ⟨𝐺^𝑢, 𝐺^𝑣⟩  where  𝐺^𝑢, 𝐺^𝑣  are now weighted Gram matrices given by  𝐺^𝑢 = 1/𝑛 ∑_𝑖 𝑎_𝑖 𝑢_𝑖⊗𝑢_𝑖  and similarly for  𝐺^𝑣 . One can then apply SAGram/SOGram to the weighted Gramians.
>
> 3) It is our intention to open-source our TensorFlow implementation in the near future.
>
> [1] Behnam Neyshabur and Nathan Srebro. On symmetric and asymmetric lshs for inner product search. In Proceedings of the 32nd International Conference on Machine Learning (ICML 2015).
> [2] Anshumali Shrivastava and Ping Li. Asymmetric lsh (alsh) for sublinear time maximum inner product search (mips). In Proceedings of the 27th International Conference on Neural Information Processing Systems (NIPS 2014).
> [3] Xu Zhang, Felix X. Yu, Sanjiv Kumar, and Shih-Fu Chang. Learning spread-out local feature descriptors. In IEEE International Conference on Computer Vision (ICCV 2017).

---

### Public Comment · (anonymous) · 2018-10-13
**Good paper**

.
I have several questions:

first, is using whole data or whole unobserved data necessary? Using whole data is better than sampling methods? I think it may depend, for some relative dense data such as in nlp-word embedding task, particularly for large word corpus, using whole data performs much worse than the sampling methods.  The performance of whole data based models is largely determined by the weighting of the unobserved or negative  examples. for example in [Bayer et al., 2017], they only use a constant weight and compare with very simple baseline. The model is not applicable for models with weights that associate with both users and items in the recommendation scenario. it is unknown whether whole data based method can beat state-of-the-art. do authors agree？

2 The model structure is limited to the dot product structure. Although it is a very popular structure in previous literature , it is not the case for deep models. A simple dot product structure is limited in modeling complicated relations. The common way is to add a full-connected layer on top of dot product. it seems that the current model does not support this popular structure.

3 the current optimization method is  limited to least square loss? what about logistic loss for classification

4 The mathematical derivation in section2.2 is very hard to follow. Can you give some motivations and a little bit more details.

5 what about the negative weighting design in equation 1?
6 eq.(1) is  not  clear ? why the first term is \sum_i^n as the number of observed examples should be much larger than n
why the second term is \sum_i^n\sum_i^n, e,g, in recommender system, the number of user and items are different.
6 will you release the code if it is accepted. The mathematics are kinda very hard to follow for most readers. Do you think the algorithm is good to be used in industry?

---

> ### Author Response · Authors · 2018-10-14
> **Thank you for your comments**
>
> Thank you for your comments. We will discuss each point below.Thank you for your comments. We will discuss each point below.
>
> 1) We agree that it is often a good idea to use non-uniform weights, (as well as non-uniform sampling distributions), and the proposed methods support these variants. We did not discuss non-uniform weights to avoid overloading the presentation, but we can certainly add a section to the appendix. As discussed in our previous comment, if we define the penalty as 1/𝑛^2 ∑_𝑖 ∑_𝑗 𝑎_𝑖 𝑏_𝑗 ⟨𝑢_𝑖, 𝑣_𝑗⟩^2 , (where in a recommendation setting, 𝑎_𝑖 is a user-specific weight and 𝑏_𝑗 is an item-specific weight), then this expression is equivalent to ⟨𝐺^𝑢, 𝐺^𝑣⟩ where 𝐺^𝑢, 𝐺^𝑣 are weighted Gram matrices, defined by 𝐺^𝑢 = 1/𝑛 ∑_𝑖 𝑎_𝑖 𝑢_𝑖⊗𝑢_𝑖 and similarly for 𝐺^𝑣. The same methods (SAGram, SOGram) can be applied to the weighted Gramians.
>
> 2) The dot product structure remains important in recent literature, e.g. [1, 2, 3], especially in retrieval settings where one needs to score a large corpus, as finding the top-k items in a dot product model is efficient (see literature on Maximum Inner Product Search, e.g. [4, 5] and references therein). In addition to such models, our methods can also apply to arbitrary models using the Global Orthogonal regularizer described in [6]. The effect of the regularizer is to spread-out the distribution of embeddings, which can improve generalization. We show in Appendix C that this regularizer can be written using Gramians, thus one can apply SOGram or SAGram to such models.
>
> 3) On the choice of loss function: the loss on observed pairs (the function ℓ in our notation) is not limited to square loss, and could be logistic loss for example. The penalty function on all pairs, (𝑔 in our notation) is a quadratic function. It can be extended to a larger family (the spherical family discussed in [7]), but this is beyond the scope of this paper.
>
> 4) On the derivation of the Gramian formulation: we gave a concise derivation in Section 2.2 due to space limitations, but we can expand here and give some intuition. The penalty term 𝑔 is a double-sum 1/𝑛^2 ∑_𝑖 ∑_𝑗 ⟨𝑢_𝑖, 𝑣_𝑗⟩^2 . If we focus on the contribution of a single left embedding 𝑢_𝑖 , we can observe that this is a quadratic function 𝑢 ↦ ∑_𝑗 ⟨𝑢, 𝑣_𝑗⟩^2 . Importantly, this is the same quadratic function that applies to all the 𝑢_𝑖 (independent of 𝑖 ). A quadratic function on ℝ^𝑑 can be represented compactly using a 𝑘×𝑘 matrix, and this is exactly the role of the Gramian 𝐺^𝑣, and because the same function applies to all 𝑢_𝑖, we can maintain a single estimate and reuse it across batches (unlike sampling-based methods that recompute the estimate at each step). There is additional discussion in Appendix C on the interpretation of this term.
>
> 5) On the choice of the weight 𝜆: as mentioned in the experiments, this is a hyper-parameter that we tuned using cross-validation. Intuitively, a larger 𝜆 puts more emphasis on penalizing deviations from the prior, while a lower 𝜆 emphasizes fitting the observations. We have experiments in Appendix C that explore this effect, e.g. the impact on the embedding distribution in Figure 4, and the impact on precision in Figure 5.
>
> 6) In eq (1), 𝑛 denotes the number of observed pairs (size of the training set). To simplify, we also define the Gramians as a sum over training examples, although in a recommendation setting, this can be rewritten as a sum over distinct users and distinct items. More precisely, if we let S be the set of users, and 𝑓_s the fraction of training examples which involve user s, then 𝐺^𝑢=1/𝑛 ∑_𝑖 𝑢_𝑖⊗𝑢_𝑖 = ∑_s∈S 𝑓_s 𝑢_s⊗𝑢_s.
>
> 7) We plan to open-source our TensorFlow implementation in the near future.
>
> [1] P. Neculoiu, M. Versteegh and M. Rotaru. Learning Text Similarity with Siamese Recurrent Networks. Proceedings of the 54th Annual Meeting of the Association for Computational Linguistics, 2016.
> [2] M. Volkovs, G. Yu, T. Poutanen. DropoutNet: Addressing Cold Start in Recommender Systems. NIPS 2017.
> [3] P. Covington, J. Adams, E. Sargin. Deep Neural Networks for YouTube Recommendations. Proceedings of the 10th ACM Conference on Recommender Systems (RecSys 2016).
> [4] B. Neyshabur and N. Srebro. On symmetric and asymmetric lshs for inner product search. ICML 2015.
> [5] A. Shrivastava and P. Li. Asymmetric lsh (alsh) for sublinear time maximum inner product search (mips). NIPS 2014.
> [6] X. Zhang, F. X. Yu, S. Kumar, and S. Chang. Learning spread-out local feature descriptors. In IEEE International Conference on Computer Vision (ICCV 2017).
> [7] P. Vincent, A. de Brebisson, and X. Bouthillier. Efficient exact gradient update for training deep networks with very large sparse targets. In NIPS 2015.

---

> > ### Public Comment · (anonymous) · 2018-10-20
> > **thanks for your reply!**
> >
> > thanks for your explanation. no doubt, this is an excellent work.  I just read the answer for my first question (will read others  later). When talking about the weight setting, I mean a_ij which involves both users and items, not only user-specific or item-specifc weight. a_ij is very common and it seems that  it cannot always be rewritten as a_i b_j. Do the algorithm apply in this settting?
> > 2 I still think the dot product structure in fig1 is not that popular recently, kinda of a bit popular  when deep learning is just in the starting stage. Do you find this structure much better than a basic factorization machines（ just a digression.
> > Btw: what do you think applying this algorithm in industry :)

---

> > > ### Author Response · Authors · 2018-10-26
> > > **On using more general weight matrices**
> > >
> > > 1) For observed pairs, one can use arbitrary weights  𝑤_𝑖𝑗 . For the unobserved data, in our problem setting, the set of all possible pairs (i, j) is too large to specify an arbitrary weight matrix (say if the vocabulary size is 10^7 or more, the full weight matrix would have more than 10^14 entries). In such situations one needs to provide a concise description of this weight matrix. One such representation is the sum of a sparse + low-rank component, and our methods handle this case: the sparse component can be optimized directly, and the low-rank component can be optimized using our Gramian estimation methods. The previous answer describes the rank-1 case where  𝑤_𝑖𝑗 = 𝑎_𝑖 𝑏_𝑗 , and the same argument generalizes to the low-rank case (for a rank-r matrix weight matrix, one needs to maintain 2*r Gramians).
> > >
> > > 2) In retrieval setting with a very large corpus, the dot product structure can be the only viable option, as scoring all candidates in linear time is prohibitively expensive, while maximum inner-product search is approximated in sublinear time. As mentioned above, even in models that don't have the dot product structure, our method applies to the global orthogonal regularizer in any embedding layer.
> > > We believe our methods are applicable to industrial settings. Our experiments suggest that the relative improvement (w.r.t. existing sampling based methods) grows with the corpus size (see Table 2), so we expect to see large improvements in applications with very large corpora. As for comparing different model classes (neural embedding models Vs. factorization machines) this is outside the scope of the paper, our focus is instead on developing efficient optimization methods for the neural embedding model class.

---

### Author Response · Authors · 2018-11-16
**Thank you for the comments; revision uploaded**

We would like to thank all reviewers for their careful reading and helpful suggestions. We have uploaded a revision of the paper with the following changes:
- We added a new section to the appendix (Appendix C) discussing how to adapt the methods to a non-uniform weight matrix.
- We added Appendix E.1 to relate the gradient estimation error to the Gramian estimation error, with a numerical experiment (Figure 6) showing the effect of our methods on gradient estimates.
- We added a comment to the conclusion to emphasize that our experiments were focused on problems with very large vocabulary size.
- We rearranged the introduction, and improved transitions between sections.
- We added comments to the numerical experiments (Section 4 and Appendix E) highlighting the effect of the batch size and of the sampling distribution.

We thank the reviewers again for their time and helpful comments.

---

### Meta-Review · Area_Chair1 · 2018-12-16
**Good paper on fast stochastic learning of embedding models.**

**Confidence:** 5
**Recommendation:** Accept (Poster)

**Metareview:**

This paper presents methods to scale learning of embedding models estimated using neural networks. The main idea is to work with Gram matrices whose sizes depend on the length of the embedding. Building upon existing works like SAG algorithm, the paper proposes two new stochastic methods for learning using stochastic estimates of Gram matrices.

Reviewers find the paper interesting and useful, although have given many suggestions to improve the presentation and experiments. For this reason, I recommend to accept this paper.

A small note: SAG algorithm was originally proposed in 2013. The paper only cites the 2017 version. Please include the 2013 version as well.